# Autophagy Function and Regulation in Kidney Disease

**DOI:** 10.3390/biom10010100

**Published:** 2020-01-07

**Authors:** Gur P. Kaushal, Kiran Chandrashekar, Luis A. Juncos, Sudhir V. Shah

**Affiliations:** 1Renal Section, Central Arkansas Veterans Healthcare System Little Rock, Arkansas and Division of Nephrology, 4300 W 7th St, Little Rock, AR 72205, USA; LJuncos@uams.edu (L.A.J.); Shahsudhirv@uams.edu (S.V.S.); 2Department of Internal Medicine, University of Arkansas for Medical Sciences, 4301 W. Markham, Little Rock, AR 72205, USA; Kchandrashekar@uams.edu

**Keywords:** autophagy, acute kidney injury, chronic kidney disease, diabetic nephropathy, mTORC1, AMPK, renal fibrosis, apoptosis, regulated necrosis

## Abstract

Autophagy is a dynamic process by which intracellular damaged macromolecules and organelles are degraded and recycled for the synthesis of new cellular components. Basal autophagy in the kidney acts as a quality control system and is vital for cellular metabolic and organelle homeostasis. Under pathological conditions, autophagy facilitates cellular adaptation; however, activation of autophagy in response to renal injury may be insufficient to provide protection, especially under dysregulated conditions. Kidney-specific deletion of *Atg* genes in mice has consistently demonstrated worsened acute kidney injury (AKI) outcomes supporting the notion of a pro-survival role of autophagy. Recent studies have also begun to unfold the role of autophagy in progressive renal disease and subsequent fibrosis. Autophagy also influences tubular cell death in renal injury. In this review, we reported the current understanding of autophagy regulation and its role in the pathogenesis of renal injury. In particular, the classic mammalian target of rapamycin (mTOR)-dependent signaling pathway and other mTOR-independent alternative signaling pathways of autophagy regulation were described. Finally, we summarized the impact of autophagy activation on different forms of cell death, including apoptosis and regulated necrosis, associated with the pathophysiology of renal injury. Understanding the regulatory mechanisms of autophagy would identify important targets for therapeutic approaches.

## 1. Introduction

Recent discoveries elucidating the molecular machinery of autophagy and its fundamental role in important cellular functions have demonstrated the relevance of autophagy in human diseases. The Nobel Committee for Physiology or Medicine recognized Yoshinori Ohsumi for his novel work in identifying the biological process of autophagy and defining its critical function [1], awarding him the Nobel prize in 2016. Therefore, autophagy research on the pathogenesis of numerous diseases, including renal injury, is an area of great interest that may provide valuable insights into potential therapeutic opportunities. 

The process of autophagy involves the formation of a double-membrane structure known as an autophagosome, which first sequesters the cellular constituents and subsequently delivers them to the lysosome for degradation [2] (Figure 1). This dynamic process culminates in the recycling of degraded products for the biosynthesis of new cellular components and for meeting the energy needs of the cell [2]. Autophagy-lysosomal and ubiquitin-proteasomal systems are the two major evolutionarily conserved cellular degradation pathways that play pivotal roles in regulating cellular homeostasis and maintaining quality control mechanisms [3,4]. The ubiquitin-proteasomal pathway involves a multimeric proteasome for the degradation of ubiquitinated small and short-lived proteins, whereas the autophagy process uses lysosomal hydrolases to degrade large intracellular heterogeneous misfolded proteins, protein aggregates, and damaged macromolecules, and has the ability to degrade entire damaged organelles and invading microorganisms [3]. The details of the autophagy process are summarized in Figure 1.

## 2. Basal Autophagy in the Kidney and Other Organs

A basal level of autophagy fulfills a vital role in cellular metabolism and organelle homeostasis by degrading and recycling damaged proteins, macromolecules, and organelles. The importance of basal autophagy in cellular homeostasis in the kidney and other organs has been demonstrated in conditional autophagy-knockout (KO) mice (tissue-specific Atg5- or Atg7-KO mice) and autophagy-deficient cells. The accumulation of protein aggregates and inclusion bodies has been observed in autophagy-deficient renal tubular cells [5,6], hepatocytes [7,8], cardiomyocytes [9], and neural cells [10,11]. In addition, proximal tubule-specific autophagy-KO mice accumulated deformed or damaged mitochondria, p62, ubiquitin-positive inclusion bodies, and misfolded protein aggregates; and they exhibited increased proximal tubule cell apoptosis [5]. These mice at 24 months of age exhibited multiple characteristics of cellular senescence, including renal function loss, mitochondrial damage, nuclear DNA damage, and fibrosis [6], compared to wild-type mice. Moreover, podocyte-specific Atg5-deficient mice were found to develop mild albuminuria, podocyte loss, late-onset glomerulosclerosis, accumulation of oxidized and ubiquitinated protein aggregates, endoplasmic reticulum (ER) stress, and proteinuria [12]. Altogether, these findings indicate the pivotal role of basal autophagy in cellular remodeling and intracellular homeostasis in both tubules and podocytes. 

## 3. Autophagy in Response to Stress and Kidney Injury

Autophagy activation is an adaptive response to a wide variety of internal and external cellular stresses (e.g., cell starvation, hypoxia, nutrient and growth factor deprivation, oxidant injury, genotoxic agents, and other damaging insults) as well as pathological conditions. Stress-induced autophagy can support cell survival by eliminating and recycling damaged macromolecules, protein aggregates, and organelles [13,14,15,16,17]. Furthermore, in response to stress, organelle-specific autophagy, known as selective autophagy, eliminates and recycles damaged organelles, including mitochondria, peroxisomes, lysosomes, ER, and even the nucleus [18]. Although stress-induced autophagy is considered to be primarily protective [15], its ultimate effect on cell survival may depend on the relative activation and completion of the autophagy process. Ideally, optimal autophagic flux ensures an autophagy-mediated supply of bioenergy molecules (e.g., ATP) and break down products (e.g., amino acids and fatty acids) to maintain cellular biosynthesis. The pro-survival function may depend on the autophagic suppression of the stress-mediated cell death pathways, including apoptosis and regulated necrosis [19,20,21]. 

Indeed, dysregulation or failure of the autophagy pathway or mutations in the autophagy-related genes (*Atg* genes) results in various human pathologies, including cancer, neurodegenerative diseases, chronic inflammatory diseases, and cardiac failure [22,23,24]. Autophagy may also promote cell death under some special circumstances. It has been suggested that high levels of autophagy may cause excessive digestion of cellular constituents, resulting in cell death. For example, a high level of autophagy induction by the cell-permeable peptide transactivator of transcription (TAT)-beclin-1 derived from beclin-1 in cell cultures causes cell death [25]. Autophagic cell death, called ‘autosis’, is a nonapoptotic cell death mechanism triggered by hypoxia, starvation, or cell-permeable beclin-1-derived autophagy-inducing peptides and is regulated by the Na-K-ATPase pump [25]. Moreover, cell death by autophagy is promoted by reactive oxygen species produced upon degradation of ferritin by autophagy, a process known as ferroptosis [26]. 

Renal tubular epithelial cells under injury conditions are exposed to multiple stresses, including oxidative stress, hypoxia, nutrient and energy depletion, endoplasmic reticulum (ER) stress, mitochondrial damage, and genotoxic stress, all of which can activate autophagy. However, insufficient or defective autophagy due to impaired clearance of damaged macromolecules and organelles is unable to provide protection from cellular stress in acute kidney injury (AKI) and other renal diseases. The specific role of autophagy in models of AKI and progressive renal disease has been revealed by using both pharmacological and genetic approaches (described below).

### 3.1. Autophagy in AKI 

Autophagy is activated in the kidney in AKI induced by ischemia-reperfusion (IR), cisplatin, and sepsis. The role of autophagy in AKI using both genetic and pharmacological approaches has been recently reviewed [27,28,29]. Conditional proximal tubule-specific *Atg5*- or *Atg7*-KO mice subjected to IR, cisplatin, or sepsis-induced AKI have consistently demonstrated worsened outcomes compared to wild-type mice, supporting a pro-survival role of autophagy in models of AKI [27,28]. Compared to control mice, proximal tubule-specific autophagy-deficient mice exhibited increased tubular damage, loss of renal function, tubular cell apoptosis, mitochondrial damage, and accumulation of p62 and ubiquitin-positive inclusion bodies in response to IR [5,6,30,31]. However, mice with conditional selective deletion of *Atg5* from the proximal tubular S3 segment exhibited a sharp rise in cell death (TUNEL positive cells but no increase in caspase-3 activation) at 2 h after IR, but less tubular damage and inflammation 3 days later compared to normal mice [32]. Hence, the outcome of IR injury differs depending on whether *Atg5* is deleted from the S3 segment alone versus from all three segments (S1, S2, and S3) of the tubule [5,30]. An increase in the TUNEL positive tubular cells with an increase of caspase-3 activity in mice deficient in *Atg5* in all segments, as well as in mice with *Atg5* deleted in the S3 segment alone without caspase-3 activation, suggests the involvement of different pathways of cell death [33]. Different modes of cell death, including apoptosis and regulated necrosis (necroptosis, ferroptosis, and parthanatos as described below), have been recently reported to occur during AKI [27,34]. Since autophagy inhibition by pharmacological approaches activates cell death pathways in renal [27,35], as well as in non-renal cells [19,36], the pro-survival effect of autophagy activation should then influence the interplay between autophagy and different cell death pathways and influence the cell fate.

### 3.2. Autophagy in Renal Interstitial Fibrosis and Progressive Kidney Disease

A hallmark of chronic kidney disease (CKD) is a progressive deposition of extracellular matrix proteins, which correlate well with the deterioration of renal function, regardless of the etiology of the primary insult [37,38,39]. In addition to various causes of CKD, acute kidney injury (AKI) is also a major contributing factor in the progression of CKD due to abnormal post-AKI recovery and ensuing progressive fibrosis, leading to end-stage renal disease (ESRD) [40,41].

To determine the role of autophagy in renal fibrosis, most studies have used the unilateral ureteral obstruction (UUO) model [42]; this model exhibits time-dependent induction of autophagy accompanied by tubular atrophy, tubular cell death, and interstitial fibrosis [43,44]. The autophagy inhibitor 3-methyladenine (3-MA) enhanced tubular apoptosis and interstitial fibrosis in obstructed kidneys [44]. In addition, transgenic mice with heterozygous deletion of beclin-1 (beclin-1 ± mice) showed increased deposition of type-1 collagen [45]. LC3)-KO and beclin-1 ± mice subjected to the UUO model revealed increased deposition of collagen accompanied by increased levels of transforming growth factor (TGF)-β1 in the obstructed kidney [46]. In this model, the induction of autophagy in distal tubular epithelial cells afforded protection from renal tubulointerstitial fibrosis through the regulation of the TGF-β/signal transducers and transcriptional modulator (Smad)4 signaling pathway and NOD-LRR and pyrin domain-containing protein3 (NLRP3) inflammasome/caspase-1/interleukin-1 β (IL-1β) signaling pathway [47]. Conditional deletion of *Atg5* enhanced renal interstitial fibrosis and promoted cell cycle arrest at G2/M [48]. It was shown that the deletion of *ATG5* in proximal tubular epithelial cells promoted leukocyte infiltration and expression of proinflammatory cytokines, while overexpression of ATG5 inhibited the inflammatory response in an autophagy-dependent manner via blocking nuclear factor kappa-light chain enhancer of activated B cells (NF-κB) signaling that provides protection against renal inflammation and accompanied fibrosis [49]. 

A recent study showed that phosphatase and tension homologue (PTEN)-induced kinase 1/ (PINK1)/mitofusion 2 (MFN2)/Parkin-mediated macrophage mitophagy is downregulated during kidney fibrosis, and loss of either Pink1 or Parkin promoted macrophage development toward profibrotic/M2 macrophages and subsequent renal fibrosis [50]. Additionally, autophagy induced by the histone deacetylase inhibitor, valproic acid, suppressed renal fibrosis in mice subjected to UUO [51]. Taken together, these studies support that autophagy suppresses renal fibrosis in obstructed kidneys and may provide a pro-survival role (Table 1). Rubicon, a negative regulator of autophagy, increased during aging, suppressed autophagic activity, and caused fibrosis in mouse kidney and α-Syn accumulation in mouse brain [52]. In streptozotocin (STZ)-induced diabetic nephropathy in rats, microRNA (miR)-22 upregulation was associated with increased fibrosis and suppression of autophagy [53]. In normal rat kidney (NRK)-52E cells, rapamycin-induced autophagy reduced high glucose-induced collagen IV (Col IV), and *α*–smooth muscle actin (*α*-SMA) expression and overexpression of miR-22 suppressed autophagic flux and induced the expression of Col IV and *α*-SMA [53]. Also, triptolide (TP), a traditional Chinese medicine, reduced fibrosis by increasing autophagy via the miR-141-3p/PTEN/protein kinase B (Akt)/mTOR pathway in STZ-induced diabetic nephropathy in high fat diet (HFD)-fed rats [54]. 1,25-dihydroxyvitamin D3 ameliorated Ang II-induced tubulointerstitial fibrosis, expanded mesangial regions and foot process fusion, and impaired autophagy by improving mitochondrial dysfunction and by modulating autophagy [55]. Elafibranor, a novel dual peroxisome proliferator-activator receptor *α*/*δ* (PPAR*α*/*δ*) agonist, protected HFD mice with CKD by improving kidney-specific protective effects, including preservation of glomerular/tubular barrier protein, maintenance of the structure, antioxidative stress, and activation of sirtuin (SIRT)-autophagy [56]. Postconditioning (POC) following IR injury reduced renal damage and renal fibrosis by increased autophagy [57]. *Periostin* gene*,* also known as osteoblast-specific factor-2 that plays a role as a profibrotic and proinflammatory factor [58], is upregulated in kidneys with 5-6 nephrectomy and impaired autophagy flux. Knockdown of *periostin* afforded protection against 5/6 nephrectomy-induced intrarenal renin-angiotensin system activation, fibrosis, inflammation in rats, and improved autophagy flux [59], suggesting that *periostin*-induced impaired autophagy is involved in the inflammation and fibrosis in the profibrotic model. In contrast, other studies have recently shown that the induction of autophagy results in renal fibrosis. Persistent activation of autophagy during UUO promoted renal interstitial fibrosis, macrophage infiltration, and tubular atrophy [60]. Proximal tubule-specific deletion of *Atg7* suppressed tubular atrophy, nephron loss, interstitial macrophage infiltration, interstitial fibrosis, and expression of the profibrotic factor fibroblast growth factor 2 (FGF2) [60]. Furthermore, a Chinese herb rhubarb and its bioactive component rhein that improved renal function and the glomerular filtration rate (GFR) in stage 3 and 4 patients with CKD [61] inhibited autophagy. Protein kinase Cα (PKCα) is activated during UUO fibrotic kidney, and inhibition of PKCα blocked autophagic flux in fibroblasts of the fibrotic kidneys and prevented fibroblast activation and kidney fibrosis [62]. Rhubarb also suppressed renal fibrosis [63]. These studies are summarized in Table 1. In view of the above results, additional studies with different experimental models of renal fibrosis are required using both genetic and pharmacological approaches to better understand the definitive role of autophagy in renal interstitial fibrosis.

## 4. Regulation of Autophagy by mTORC1-Dependent and Alternative mTORC1-Independent Pathways in Renal Injury

Accumulating evidence indicates that autophagy is predominantly regulated by the mammalian target of rapamycin (mTORC1)-dependent signaling pathway; however, recent studies have demonstrated alternative mTOR-independent pathways involved in regulating autophagy (Figure 1 and Figure 2) [13,15,17,18,64,65,66]. 

### 4.1. mTORC1-Dependent Autophagy Regulation

#### 4.1.1. mTORC1-ULK1 Pathway

The interaction of the serine-threonine kinase mTORC1 with autophagy is mediated by the mTOR-ULK1/2 signaling pathway (Figure 1) [67,68,69]. mTORC1 senses and responds to diverse stress conditions, including metabolic/energetic stresses, oxidative stress, ER stress, and genotoxic stress, and it is inactivated by most if not all of these stresses [70,71]. Suppression of mTORC1 activates autophagy, and therefore, mTORC1 is considered a negative regulator of autophagy. Inactivation of mTORC1 activates the ULK1 complex of the autophagy pathway by dephosphorylation of ULK1 at the Ser-757 site, which then permits ULK1 to phosphorylate its associated components Atg13 and FIP 200, resulting in ULK1 activation. Active ULK1 then activates downstream targets of the autophagy pathway, including members of the vacuolar protein sorting 34 (VPS34) complex1, which promotes the synthesis of phosphatidylinositol 3-phosphate (PI3P) synthesis [72,73], a key player in membrane dynamics involved with the autophagosome preassembly machinery. In contrast, the activation of mTORC1 in response to growth factors, nutrients, and increased energy levels promotes anabolic processes (e.g., protein synthesis, lipogenesis, cell growth, metabolism, and proliferation) and suppresses catabolic processes, including autophagy [70,74,75]. mTORC1 represses autophagy upon inhibiting the kinase activity of ULK1/2 by phosphorylation of ULK1 (Ser-638/758) and its associated partner Atg13 (Ser-258) [76]. Thus, mTORC1 plays a pivotal role in the induction and regulation of autophagy.

#### 4.1.2. mTORC1-ULK1-Mediated Autophagy in Renal Injury

mTORC1 plays an important role in maintaining renal tubular homeostasis. It has been demonstrated that tubular-specific mTORC1-deletion leads to an abnormal urine concentration, tubular cell loss, impaired mitochondrial biogenesis, metabolic and transport dysfunctions, and slow progression to renal fibrosis [77]. In an IR model, mTORC1-deficient mice exhibited greater tubular damage and pronounced apoptosis [77]. Moreover, the upregulation of mTOR activity has been documented in the renal IR model of AKI [78], a rat model of subtotal nephrectomy [79], glomerular injury, including diabetic nephropathy [80,81], and polycystic kidney disease [82]. In an IR model, mTORC1-deficient mice exhibited more tubular damage, pronounced apoptosis, and delayed recovery relative to wild-type mice [77]. Furthermore, inhibition of mTOR with rapamycin or its analog in renal IR caused a delay in functional recovery [83,84] due to the induction of apoptosis and inhibition of tubular cell proliferation [78]. 

**Table 2 biomolecules-10-00100-t002:** mTORC1-dependent autophagy regulation in kidney disease.

mTORC1-ULK1 Pathway in Kidney			
Model/ Kidney Disease	Agent/Drug	Effect on Autophagy	Reference
Murine model of sepsis	Temsirolimus	↑ autophagy via inhibition of mTORC1	[85]
	AMPK activator AICAR		
Adriamycin-induced podocyte injury	Rapamycin and podocyte-specific Atg7 KO	↑ autophagy	[86]
G1 and G2 generic variants in APOL1 gene in podocytes		↓ autophagic flux by miR-193a mediated mTOR inhibition	[87]
High glucose-induced podocyte injury	Ursolic acid	↑ autophagy and improves podocyte injury	[88]
db/db mouse model	AGEs	↓ podocyte autophagy via activation of mTORC1 and inhibition of nuclear translocation of TFEB	[89]
	Pyridoxamine	↑ autophagic flux	
kkAy mice with DN	Apelin	↓ autophagic flux and promotes podocyte injury and progression of DN	[90]
High glucose-induced podocyte injury	Notoginsenoside R1	↑ autophagy	[91]
MRL*^lpr/lpr^*mice model of LN	Rapamycin	↑ autophagy	[92]
Renal transplant	Sirolimus	↑ autophagy	[93]
**AMPK-mTORC1/ULK1 pathway in Kidney**			
Renal IR injury	AMPK activators (AICAR, metformin)	↑ autophagy	[94]
	Quercetin, omega 3-PUFA and pioglitazone	↑ autophagy	[94,95,96,97]
Cisplatin nephrotoxicity	Metformin and neferine	↑ autophagy	[98,99]
Sepsis by CLP	SIRT3	↑ autophagy	[100]
STZ induced DN	Saponin astragaloside and mangiferin	↑ autophagy	[101,102]
db/db diabetic nephropathy	Cincalcet (type II agonist of calcium sensing receptor)	↑ autophagy	[103]
	HO-1, berberine and progranulin	↑ autophagy	[104,105,106]
HFD	AICAR	↑ autophagy	[107]
	Fenofibrate	↑ autophagy	[108,109]

In a murine model of sepsis, autophagy activation by either a mTORC1 inhibitor temsirolimus or an AMPK activator 5-aminoimadazole-4-carboxamide ribonucleotide (AICAR) was protective against the development of sepsis-associated AKI [85]. Additionally, the upregulation of mTORC1 activity in both the tubular and glomerular epithelium in human native kidney biopsies was associated with increased levels of serum creatinine in native kidney disease, and elevated mTORC1 levels were found to be accompanied by interstitial fibrosis and acute cellular rejection in transplant kidney biopsies [110]. 

In cultured podocytes, adriamycin-induced podocyte injury was prevented by activation of autophagy with rapamycin, and in inducible podocyte-specific Atg7 knockout (Podo-Atg7-KO) mice, podocyte injury was enhanced compared to control mice [86]. The G1 and G2 generic variants in the *APOL1* gene in podocytes, involved in an increased risk of developing chronic kidney disease in African Americans, were able to suppress podocyte autophagy flux [111]. Further studies showed that APOL1 risk alleles diminished autophagy flux in podocytes not only by miR193a-mediated mTOR inhibition but also by suppression of phosphatidylinositol -3-kinase regulatory subunit 3 (PIK3R3) transcription and increased expression of rubicon [87]. Podocyte injury due to high glucose reduced podocyte autophagy and treatment with ursolic acid improved podocyte injury through miR-21 inhibition and PTEN expression, which suppressed PI3K/Akt/mTOR pathway and restored autophagy [88]. In the *db/db* mouse model of diabetes mellitus, high levels of advanced glycation end-products (AGEs) and high glucose suppressed podocyte autophagy by activating mTORC1 and inhibiting the nuclear translocation of transcription factor EB (TFEB) [89]. Inhibition of AGE formation by pyridoxamine restored autophagic flux. An adipokine, apelin, that promoted podocyte injury and the progression of diabetic nephropathy in kkAy mice, inhibited autophagy in podocytes through activating the mTOR pathway [90]. A Chinese herbal compound notoginsenoside R1 (NR1), a major component of *Panax notoginseng*, increased mTORC1-dependent autophagy in podocytes exposed to high glucose and protected podocyte injury [91]. Podocyte-selective mTOR-KO mice worsened proteinuria compared to control littermates, and podocytes from these mice accumulated autophagosomes, autolysosomes, and damaged mitochondria [112]. Treatment of human podocytes with mTOR inhibitor rapamycin also accumulated autophagosomes and autophagolysosomes. These studies suggest that disruption of the autophagic pathway may be involved in the pathogenesis of proteinuria following treatment with mTOR inhibitors [112]. In the MRL*^lpr/lpr^*mice model of lupus nephritis (LN), autophagy induction by rapamycin afforded protection in podocyte from IgG-LN-induced apoptosis and interferon (IFN)-α-induced derangement of podocin [92]. In biopsies of sirolimus-treated renal transplant patients, mTOR inhibitor, sirolimus, significantly increased autophagosome formation, as shown [93]. This study may suggest increased autophagosomal accumulation rather than increased autophagic flux. The mTORC1-ULK1 pathway involved in autophagy regulation in renal diseases is summarized in Table 2. It should be emphasized that mTORC1 is a multifunctional protein with a regulatory function in protein synthesis, nucleotide synthesis, lipid synthesis, mitochondrial biogenesis, and autophagy suppression (Figure 1), and therefore, suppression of mTOR may impact the biosynthetic activity of the cell.

#### 4.1.3. AMPK-mTORC1/ULK1 Pathway 

AMP-activated protein kinase (AMPK) is activated in response to a high AMP/ATP ratio in the cell [113]. This energy sensitive enzyme plays a key role in regulating cellular energy metabolism. AMPK activates the autophagy pathway by suppressing mTORC1 [68], as well as by phosphorylating ULK1 at multiple sites (Figure 1). Inhibition of mTORC1 by AMPK is achieved by AMPK-mediated phosphorylation of tuberous sclerosis complex 2 (TSC2), thereby increasing TSC1/2 activity, which ultimately represses mTORC1 [114,115]. Also, AMPK can directly phosphorylate ULK1 at Ser-317 and Ser-777 in the ULK1/2 complex that activates the autophagy pathway [116,117]. Phosphorylation of other ULK1 sites by AMPK regulates the localization of the Atg 9A protein, which is involved in the organization of the preautophagosomal/phagophore assembly site [118]. In addition, AMPK promotes autophagy activation by phosphorylating beclin-1 in the PI3 kinase catalytic subunit type 3 (PI3KC3) complex [116]. Once phosphorylated by AMPK, ULK1/2 activates PI3KC3 by phosphorylating the catalytic subunit VPS34 (Ser-249) [117], beclin-1 (Ser-15 and other sites) [72], and the PI3KC3-C1-associated protein autophagy and beclin-1 regulator 1 (AMBRA1) [119], which facilitates activation of the autophagy pathway. Moreover, ULK1 phosphorylates FUNDC1 (FUN14 domain containing 1) protein involved in the process of mitophagy [120]. Taken together, multiple pathways of AMPK-mediated activation of the ULK1/2 complex regulate autophagy activation.

#### 4.1.4. AMPK-mTORC1/ULK1-mediated Autophagy in Renal Injury

AMPK is activated under metabolic stress and plays a significant role in acute and progressive renal injury. Autophagy activation via the AMPK-mTORC1/ULK1 pathway along with the mTORC1-ULK1 pathway in renal diseases is summarized in Table 2. AMPK-mediated activation of autophagy has been shown to play a protective role against renal injury. In renal IR injury, AMPK is a key component of the cellular adaptive responses to renal ischemia. AMPK activators (5-aminoimidazol-4-carboxamide-1-*β*-d-ribofuranoside and metformin) have been shown to protect against IR, and inhibition of AMPK with compound C has been shown to worsen kidney injury [94]. Many other natural agents, including quercetin [95], omega 3-polyunsaturated fatty acid (PUFA) [96], and pioglitazone [97], afforded protection from renal IR through AMPK-mediated autophagy. Enhancing autophagy through the AMPK/mTOR-mediated pathway also provided protection against cisplatin-induced nephrotoxicity. Furthermore, metformin [98] and neferine [99], which enhance autophagy via the AMPK/mTOR pathway, ameliorate cisplatin nephrotoxicity. In a sepsis model of cecal ligation and puncture, nicotinamide adenine dinucleotide NAD-dependent protein deacetylase SIRT3, a member of the mammalian sirtuin family of proteins, provided protection from sepsis-induced AKI through AMPK/mTOR-regulated autophagy [100]. In adriamycin-treated calcium-independent phospholipase A2 (iPLA2)γ knockout (KO) mice, both AMPK activation and autophagy were increased compared with treated control mice [101]. Heme oxygenase-1 (HO-1)-mediated autophagy is AMPK dependent and has provided protection against high –glucose-induced podocyte apoptosis [102]. In models of diabetic nephropathy, many pharmaceutical agents have been shown to prevent podocyte injury via regulating AMPK/mTOR-mediated autophagy. For example, the saponin astragaloside IV (AS-IV) and the polyphenol glucoside mangiferin prevented podocyte injury through AMPK/mTOR regulated autophagy in a model of streptozotocin-induced diabetic nephropathy [103,104]. Moreover, cinacalcet, a type II agonist of calcium-sensing receptor (CaSR), ameliorated diabetic nephropathy in *db/db* mice via regulating AMPK/mTOR-mediated autophagy [105]. Other in vitro studies also have shown that heme oxygenase-1, berberine, and progranulin alleviate high glucose-induced podocyte injury through AMPK-mediated autophagy activation [102,106,107]. 

A high-fat diet (HFD) leads to lipid accumulation and decreases levels of the cellular energy sensor AMPK in the kidney, resulting in lipotoxicity. Pharmacological activation of AMPK by AICAR has been shown to prevent HFD-induced tubular cell structure impairment and associated inflammation and fibrosis [108]. Also, fenofibrate-mediated upregulation of AMPK has been demonstrated to provide protection from kidney injury consequent to HFD-induced lipotoxicity [109] or in *db/db* mice [121]. Although HFD-induced renal tubular injury is associated with impaired autophagy [122,123], it is unknown whether activation of AMPK alleviates HFD-induced dysregulation of autophagy.

#### 4.1.5. SIRT1-mTORC1 Pathway

SIRT1, a nicotinamide adenine dinucleotide (NAD)-dependent deacetylase, is another nutrient-sensing enzyme that is activated in response to an increase in the NAD/NADH ratio upon energy depletion and oxidative stress [124]. SIRT1 can stimulate autophagy by deacetylating Atg5, Atg7, and LC3 [125]. It positively regulates AMPK by deacetylating liver kinase B1 (LKB1) [126] and downregulates mTORC1 upon interaction with the TSC2 complex [127]. This pathway is likely to play an important role in renal injury and remains a fruitful area of investigation.

In addition to activation by low-energy states and nutrient deprivation, mTORC1 is inactivated in response to other stresses, including oxidative [128,129,130] and ER stress [131,132]. ER stress has been shown to induce autophagy in renal tubular cells in culture, as well as in vivo in the kidney [133,134]. Autophagy induction by mild doses of tunicamycin promotes an adaptive response and afforded protection from oxidative stress, ATP depletion, and renal ischemia-reperfusion injury (IRI) [135]. 

### 4.2. mTORC1-Independent Alternative Pathways of Autophagy Regulation in Renal Injury

The major mTOR-independent autophagy regulating pathways involve intracellular Ca^2+^ signaling, calpain, cAMP-Epac-phospholipase C-inositol 1, 4, 5-trisphosphate (PLC)-ε-IP_3_, trehalose-mediated autophagy regulation*,* transcription factor EB (TFEB)-mediated autophagy induction, forkhead box O3 (Foxo3)-mediated autophagy induction, c-Jun N-terminal kinase (JNK)-beclin-1-PI3KC3 pathways, p38, and Akt signaling (Figure 2A,B). The involvement of mTORC1-independent pathways in renal injury is summarized in Table 3.

**Figure 2 biomolecules-10-00100-f002:**
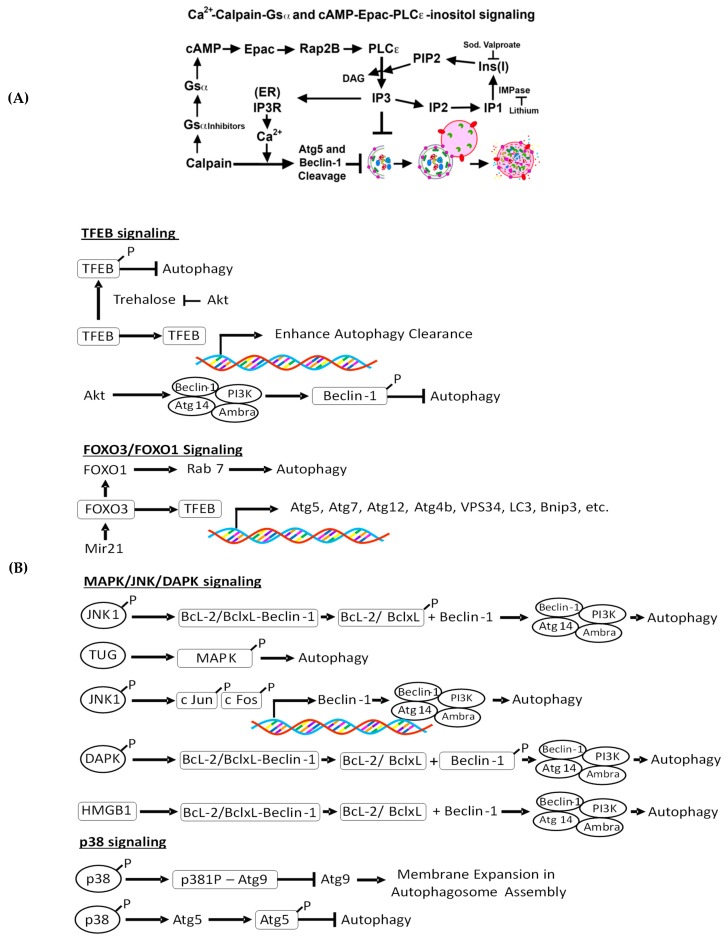
(**A**): Ca^2+^-calpain, cAMP- exchange protein directly activated by cAMP (Epac)- phosphoinositide-specific phospholipase C-ε (PLC-ε)-inositol phosphate 3 (IP_3_)_,_ and inositol signaling pathways of autophagy regulation. Calpains cleave α-subunit of G-protein (Gsα) to from active Gsα (Gsα_a_). Gsα_a_ enhances adenyl cyclase activity to generate cAMP. The cAMP also regulates autophagy via a cAMP-Epac-PLC-ε-IP_3_ pathway to generate Ins (1,4,5) P3 (IP_3_), which inhibits autophagy in an mTOR-independent manner. Also, calpains activated by cytosolic calcium cleave autophagy proteins ATG5^38,39^ and beclin-1^38^ that can inactivate autophagy. (**B**): mTORC1-independent regulation of autophagy. TFEB-mediated autophagy induction: In response to renal IR injury, TFEB is translocated to the nucleus and induces lysosomal biogenesis and autophagy. Forkhead box O3 (FoxO3)-mediated autophagy induction: FoxO3 induces activation of TFEB, which upon translocation to the nucleus, transactivates autophagy genes. Also, FoxO3 induces FoxO1, which in association with Rab 7, facilitates the fusion of the lysosome to the autophagosome. Also, janus kinase (JNK), tether containing a UBX domain for GLUT4 (TUG), death-associated protein kinase (DAPK), and p38 regulate autophagy, as shown in the figure.

#### 4.2.1. Intracellular Ca ^2+^-Mediated Autophagy Regulation and Renal Injury 

Intracellular Ca^2+^ plays an important role in the regulation of autophagy [136]. In cadmium (Cd^2+^)-induced nephrotoxicity, increased levels of Ca^2+^ due to its release from the ER inhibited autophagy in renal proximal tubules and enhanced nephrotoxicity [137] (Table 3). It has been demonstrated that increased intracellular Ca^2+^ levels prevent Rab7 recruitment to the autophagosomes that block autophagosome-lysosome fusion, resulting in autophagosome accumulation in Cd^2+^-exposed proximal tubular cells [138]. In addition, activation of the Ca^2+^-sensing receptor has been proposed to reduce Cd^2+^-induced cytotoxicity in renal proximal tubular cells [139]. Cd^2+^ induces intracellular Ca^2+^ elevation through PLC followed by stimulating p38 mitogen-activated protein kinases (MAPK) activation and suppressing extracellular signal-regulated kinase (ERK) activation, leading to increased apoptotic cell death and inhibition of cell proliferation. Cd^2+^-induced p38 activation also contributes to the autophagic flux inhibition, which aggravates Cd^2+^-induced apoptosis.

#### 4.2.2. Ca^2+^-Calpain-Mediated Autophagy Regulation and Renal Injury 

Calpains are Ca^2+^-regulated proteases [140] that can regulate autophagy in an mTOR-independent manner [141]. Calpains cleave the α-subunit of G-protein (Gsα) to from active Gsα (Gsα_a_), which enhances adenyl cyclase activity, and the resultant increase in cAMP levels inhibits autophagy [142]. Conversely, calpains activated by cytosolic calcium cleave the autophagy proteins Atg5 [143,144] and beclin-1 [143], which can inactivate autophagy. Moreover, pharmacological inhibition or genetic knockdown of calpains enhances autophagic flux without affecting mTORC1 [145]. Genetic knockdown of calpain or overexpression of the calpain inhibitor calpastatin increased autophagy and improved motor signs and delayed onset of tremors in a mouse model of Huntington disease (HD) [141]. A recent study also has shown that calpain facilitates flux of Atg9/bax-interacting factor (Bif-1) vesicles from the Golgi to the phagophore [146]. 

In cisplatin nephrotoxicity studies, the autophagy proteins beclin-1, Atg5, and Atg12 were determined to be proteolytically processed and cleaved during cisplatin injury. zVAD-fmk—a pan-caspase and calpain inhibitor—blocked the cleavage of autophagy proteins but caused impaired autophagic flux due to inhibition of lysosomal calpains [147].

#### 4.2.3. cAMP-Epac-PLC-ε-IP_3_ Pathway in Autophagy Regulation and Renal Injury 

This pathway negatively regulates autophagy independently of mTORC1 [42,43] and generates IP_3_ via the inositol pathway (Figure 2A) [145,148]. IP3, produced upon hydrolysis of phosphatidylinositol 4,5-bisphosphate (PIP_2_) by phospholipase C and free myo-inositol, inhibits autophagy. Rilmenidine, an antihypertensive drug that is approved for long-term use by the US FDA, has been demonstrated to reduce cAMP levels to promote the clearance of aggregate-prone proteins upon autophagy activation and to protect against neurodegenerative pathology in both primary neurons and a transgenic mouse model of HD [149,150]. Other IP3-reducing drugs, including lithium, carbamazepine, and valproic acid, also induce autophagy [145,150]. Lithium reduces IP_3_ and myo-inositol levels by inhibiting inositol monophosphatase, whereas carbamazepine and valproic acid inhibit the biosynthesis of inositol (Figure 2A) [142,145]. In addition, *myo*-inositol oxygenase (MIOX), a tubular cell-specific enzyme, oxidizes *myo*-inositol to d-glucuronate [151]. In the pathogenesis of diabetic kidney disease, increased expression of MIOX causes mitochondrial depolarization and fragmentation and defective autophagy, perturbing the mitochondrial homeostasis [152]. Activated MIOX during hyperglycemia induces mitochondrial fragmentation and depolarization; however, it inhibits autophagic removal of damaged mitochondria. Upon binding to its receptors, which are primary Ca^2+^ release channels in the ER, IP_3_ triggers Ca^2+^ release into the cytosol, which can inhibit or facilitate autophagy depending on the context [153]**.** IP3 receptor downregulation or selective inhibition with Xestospongin B has been reported to increase autophagy [154]. IP_3_ receptor-mediated suppression of autophagy also has been attributed to its formation of a complex with beclin-1, preventing free beclin-1 availability to promoting autophagy, and this interaction is increased or inhibited by overexpression or knockdown of Bcl-2, respectively [155]. Furthermore, IP_3_ receptor-mediated Ca^2+^ release into the cytosol can infiltrate into the mitochondria, stimulating apoptosis and inhibiting autophagy. Other studies have demonstrated that IP_3_ receptor-mediated Ca^2+^ release during starvation- [153] and rapamycin- [156] induced m-TORC1-dependent autophagy in mammalian cells. However, the increased basal autophagic flux triggered upon IP_3_ receptor inhibition appears to be independent of mTOR [136].

#### 4.2.4. c-Jun N-Terminal Kinase (JNK)-Beclin-1, p38, and Akt Signaling and Renal Injury

Other alternative pathways of mTORC1-independent autophagy regulation include the JNK-beclin-1-PI3KC3 pathways and p38, NFkB, and Akt signaling pathways (Figure 2B). Beclin-1, through its Bcl-2 homology domain, binds to Bcl-2 and inhibits autophagy under normal or nutrient-enriched conditions [157]. JNK1-mediated phosphorylation of Bcl-2 then dissociates beclin-1-Bcl2 complex under starvation, and free beclin-1 then promotes autophagy [158]. Meanwhile, death-associated protein kinase (DAPK) phosphorylates beclin-1, which dissociates the beclin-1-Bcl-xL complex to promote autophagy induction [159]. Similarly, high mobility group box 1 (HMGB1) is another pro-autophagic protein that promotes dissociation of the beclin-I-Bcl-2/Bcl-xL complex [160]. In addition, dFoxO protein, a member of the FoxO transcription factor family that regulates downstream of JNK activation [161], mediates FoxO-dependent transcription of *Atg* genes [162]. JNK1-mediated c-Jun regulates *Beclin1* transcription [163]. Trafficking of the transmembrane autophagy protein mammalian Atg9 requires p38-interacting protein (p38IP) to promote autophagy. However, p38 MAPK directly competes with mammalian Atg9 for binding of p38IP, thus, preventing Atg9 redistribution to the endosomes and the Golgi network and suppressing starvation-induced autophagy [164]. Indeed, p38 MAPK suppresses autophagy via an mTOR-independent mechanism involving Atg9-p38IP by increasing the interaction between mAtg9 and p38IP [165,166]. Moreover, active p38 phosphorylates Atg5 at threonine 75, inhibiting starvation-induced autophagy [167]. Nuclear factor-κB (NF-κB) activation has been shown to promote autophagy in various cell lines by increasing beclin-1 expression [168]. Conversely, Akt-mediated phosphorylation of beclin-1 has been determined to impair autophagy in a mTORC1-independent fashion [165] (Figure 2B).

#### 4.2.5. Trehalose-Mediated Autophagy Regulation and Renal Injury 

The nonreducing disaccharide trehalose induced autophagy in human podocytes in an mTOR-independent manner [169]. In a glomerular disease, the podocyte injury, characterized by actin cytoskeleton rearrangement and apoptosis, is ameliorated by trehalose-induced autophagy in a mTORC1-independent fashion in puromycin aminonucleoside (PAN)-treated podocytes that mimic minimal change disease, which is a common cause of nephrotic syndrome [169]. 

In addition, trehalose has been shown to provide protection from Cd^2+^-induced cytotoxicity by restoring autophagic flux in rat proximal tubular cells. In vivo bioavailability of trehalose is reduced by the enzyme trehalase, which is expressed in the kidney and intestines. Non-hydrolyzable trehalose analogs, known as lentztrehaloses [170], can be useful in vivo models of renal injury. 

In a zebrafish model of polycystic kidney disease (PKD), combination treatment with the mTOR-dependent autophagy activator rapamycin, as well as the mTOR-independent autophagy activators carbamazepine and minoxidil, markedly attenuated cyst formation and restored kidney function [171]. Furthermore, in an Akt2 KO-induced model of insulin resistance, trehalose ameliorated insulin resistance-induced kidney and skeletal muscle injury, excessive autophagy, and apoptosis [172]. Lithium chloride, an inositol monophosphatase inhibitor, has been reported to reduce IRI in many organs, such as the brain, heart, liver, and kidney [173,174,175]. Lithium chloride protects hepatocytes from IRI by inducing autophagy via modulation of both the glycogen synthase kinase 3b (GSK3b) and ERK1/2 pathways [175]. However, it is unknown whether lithium chloride-mediated protection against renal injury involves autophagy induction. Upregulation of the cAMP and MAPK/ERK1/2 pathways has been shown to result in the activation of autophagy and suppressed myocardial infarction [176]. Additionally, vitamin D-induced autophagy, shown to be regulated by MEK/ERK pathway, has been shown to protect against hepatic IR injury [177].

In a cellular model of Parkinson’s disease, nicotinamide adenine dinucleotide phosphate (NADPH) oxidase promotes Parkinsonian phenotypes-impaired autophagic flux independent of mTORC1 [178].

#### 4.2.6. TFEB-Mediated Autophagy Induction and Renal Injury

The transcription factor EB (TFEB) is a member of the basic helix-loop-helix leucine-zipper family of transcription factors and is involved in the regulation of lysosomal biogenesis [179] and autophagosome-lysosome function [180] (Figure 2B). In podocytes, Jack/Stat signaling facilitates the binding of STAT1 to the TFEB promoter and increases the expression of TFEB, thus, maintaining autophagosome-lysosome function. JACK2-deficient mice in podocytes exhibited impaired autophagy and increased urinary albumin excretion, while mice with TFEB overexpression in JAK2-deficient podocytes had restored autophagosome-lysosome function and albumin permselectivity [181]. These studies suggest that TFEB is a promising therapeutic target for improving podocyte injury in glomerular diseases. TFEB also prevented lysosomal abnormalities in cystinotic kidney cells [182], suggesting that TFEB could be targeted to treat autosomal recessive lysosomal storage disease cystinosin. In renal IR injury, TFEB was translocated to the nucleus and induced lysosomal biogenesis and autophagy. Furthermore, urolithin A treatment enhanced TFEB expression, thus, promoting autophagy and ameliorating renal IR injury [183]. In immunosuppressant tacrolimus (Tac)-induced renal injury, impaired autophagy was found to be in tubules, which exhibited FoxO3-mediated autophagy associated with a reduction of TFEB [184]. In addition, Klotho protected Tac-induced renal injury by restoring lysosomal function and improving autophagy by inducing nuclear translocation of TFEB through inhibiting phosphorylation of glycogen synthase kinase 3β (GSK3β) [184]. A recent study has demonstrated TFEB-mediated autophagic clearance of proteolipid aggregates in a mouse model of Batten disease in mTORC1-independent and Akt-dependent manners [185]. In this model, administration of the disaccharide trehalose prevented Akt-mediated TFEB phosphorylation and enabled nuclear translocation of free TFEB, thereby enhancing lysosomal function and autophagic clearance (Figure 2B, Table 3).

#### 4.2.7. MAPK/JNK/DAPK-Mediated Autophagy Induction and Kidney Injury

In a very recent study, lipopolysaccharide (LPS)-induced podocyte injury suppressed the expression of long non-coding RNA-taurine-upregulated gene 1 (TUG1), and TUG1 overexpression enhanced the level of p-MAPK/MAPK and induced autophagy, thus protecting against LPS-induced podocyte injury [186]. Additionally, in cisplatin nephrotoxicity, increased expression of TGF-β-activated kinase 1 (TAK1) resulted in phosphorylation of p38 and ERK, which enhanced excessive autophagy and caused more severe renal injury [187]. In a study related to aristolochic acid (AA)-induced nephropathy, AA treatment induced autophagy and increased ERK1/2 activity and pharmacological inhibition of ERK1/2 phosphorylation with U0126 reduced AAI-induced autophagy and increased apoptosis [188]. These studies suggest that AA induces autophagy through the ERK1/2 pathway, which may provide a mechanism for cell survival.

#### 4.2.8. FoxO3-Mediated Autophagy Induction and Kidney Disease 

The stress-responsive transcription factor forkhead box O3 (FoxO3) activates autophagy in chronic hypoxia through the transactivation of autophagy proteins and plays a protective role in kidney diseases (Figure 2B, Table 3). In a UUO model, FoxO3 was activated in the hypoxic proximal induction through the increased nuclear expression of Atg proteins, including Ulk1, beclin-1, Atg9A, Atg4B, and Bcl2/adenovirus E1B 19 kDa protein-interacting protein 3 (Bnip3) [189]. These studies provide evidence that FoxO3 is an upstream regulator of the autophagy machinery in hypoxic tubules during ureteral obstruction. Further studies focusing on the acute- to long-term responses of IRI in mouse kidneys showed that hydroxylation of FoxO3 at the proline residues was suppressed that prevented degradation of FoxO3 and promoted its nuclear accumulation and hypoxia inducing factor (HIF)-1α-mediated activation, which subsequently resulted in autophagy activation in hypoxic tubules [190]. Moreover, the deletion of FoxO3 in renal tubules aggravated the renal structural and functional damage, resulting in the development of the CKD phenotype, including increased oxidative stress and renal fibrosis during the AKI-to-CKD transition [190]. These studies support a protective role of FoxO3 during the AKI-to-CKD transition by reducing oxidative stress and increasing autophagy. The protective role of prolyl hydroxylation inhibition was also demonstrated in a study showing that the prolyl hydroxylase inhibitor roxadustat (FG-4592) afforded protection against cisplatin nephrotoxicity [191].

In studies related to ischemic preconditioning, both the expression and phosphorylation of the serine/threonine kinase, serum and glucocorticoid-induced kinase-1 (SGK1), were increased. In an in vitro model, using hypoxic preconditioning overexpression of SGK1 efficiently increased the effect in renal tubular cells [192]. Deacetylation of FoxO3a by SIRT1 promoted the increased expression of proautophagic Bcl2/adenovirus E1B 19-kDa interacting protein 3 (Bnip3) [193]. In addition, caloric restriction in aged mice protected aging kidneys from damage by increasing the expression of SIRT1 that deacetylated FoxO3a and, in turn, induced Bnip3 expression, thereby promoting mitochondrial autophagy [193]. In the human embryonic kidney cell line HEK293T and the mouse embryonic fibroblast cell line MEF, FoxO3 promoted the translocation of FoxO1 from the nucleus to the cytoplasm via AKT1-mediated phosphorylation of FoxO1 and subsequent increase in FoxO1-induced autophagy [194]. Furthermore, recent studies have shown that aldosterone plays an important role in podocyte injury. Aldosterone increases autophagy flux by upregulating FoxO1 and its effector Rab7, which facilitates the fusion of the lysosome to the autophagosome. The increased autophagy mediated by FoxO1-Rab7 has been demonstrated to provide protection against podocyte injury [195]. In a mouse model of streptozotocin-induced type 1 diabetes, lentiviral vectors mediated FoxO1 overexpression in the kidney cortex, which promoted PINK1/Parkin-mediated autophagy and thereby protected against mitochondrial dysfunction and podocyte injury [196]. Another study in dominant-negative mice showed that atrasentan increased FoxO1 expression by downregulating miR-21 and enhanced autophagy and alleviated podocyte injury by regulating the miR-21/FOXO1 axis [197]. Taken together, these studies reveal that FoxO3- and FoxO1-mediated autophagy activation affords protection against podocyte injury in diabetic nephropathy. In response to energy depletion, kidneys produce 2′,3′-cAMP, which can open mitochondrial permeability transition pores that promote autophagy through the process of mitophagy [198]. Since the renal enzyme 2′,3′-cyclic nucleotide 3′-phosphodiesterase (CNPase) can metabolize 2′,3′-cAMP to 2′-AMP, deletion of CNPase protected CNPase KO mice from IR injury due to 2′,3′-cAMP-mediated mitophagy [198].

**Table 3 biomolecules-10-00100-t003:** mTORC1-independent autophagy regulation in kidney disease.

Model/ Kidney Disease	Agent/Drug	Effect on Autophagy	Reference
Cadmium-induced nephrotoxicity	Increased calcium release from ER	↓ autophagy due to blockage in Rab7 recruitment to autophagosome	[136,137,138,139]
Cisplatin-nephrotoxicity	zVAD-fmk	↓ autophagic flux	[147]
Diabetic kidney disease	Increased expression of MIOX	↓ autophagy	[152]
	Xestospongin B	↑ autophagy	[154]
PAN-induced podocyte injury	Trehalose and analogs, known as lentztrehaloses	↑ autophagy	[169]
Zebrafish model of polycystic kidney disease	carbamazepine and minoxidil	↑ autophagy	[171]
JACK2-deficient podocyte mice	TFEB overexpression	↑ autophagy	[181]
Renal IR injury	Urolithin-A	↑ autophagy by enhanced TFEB expression	[183]
Tacrolimus-induced renal injury	Klotho	↑ autophagy by enhanced nuclear translocation of TFEB	[184]
Lipopolysaccharide (LPS)-induced podocyte injury	Overexpression of TUG1	↑ autophagy	[190]
Cisplatin nephrotoxicity	**↑****transforming growth factor-beta-activated kinase 1** (TAK1)	↑ autophagy	[191]
Aristolochic acid (AA)-induced nephropathy	AA exposure	↑ autophagy with ↑ ERK1/2 activity	[192]
UUO	FoxO3	↑ autophagy through nuclear expression of Atg proteins	[193]
Prolonged IR (AKI to CKD transition)	FoxO3	↑ autophagy	[194]
Ischemic preconditioning	Increased FoxO3 by overexpression of SGK1	↑ autophagy	[196]
Podocyte injury	Aldosterone	↑ autophagy through FoxO1-Rab7	[195]
STZ-induced DN	FoxO1 overexpression	↑ autophagy	[196]
Podocyte injury	Atrasentan	↑ autophagy by increased expression of FoxO1 by downregulating miR21	[197]

#### 4.2.9. GCN2 Kinase-Mediated Autophagy Induction and Kidney Disease

Another kinase, known as general control nonderepressible 2 (GCN2), is activated in response to deprivation of amino acids and induces autophagy [199]. Indoleamine 2, 3 dioxygenase 1 (IDO1), an enzyme involved in tryptophan metabolism, regulates GCN2 activation. In a mouse model of glomerulonephritis, IDO1-mediated activation of GCN2 kinase is required for the induction of autophagy and, ultimately, podocyte survival and suppression of inflammation [200]. In a related study in primary human renal tubular epithelial cells, it was shown that cells subjected to hypoxia in the absence of tryptophan were protected by GCN2 kinase-mediated induction of autophagy in a mtorc1-independent manner [201].

In cultured renal tubular cells, hypoxia-induced autophagy was regulated by increased expression of BNIP3 in a HIF-1α-dependent manner, whereas oxidative stress-induced autophagy was found to be dependent on increased sestrin-2 expression in a p53-dependent manner [202]. In this study, BNIP3-induced autophagosomes were localized mainly to the mitochondria, indicating that autophagy in renal cells in response to oxidative stress is mediated by mitophagy. Activation of NF-κB can also induce beclin-1 expression, showing enhanced autophagy and cell survival in multiple cell lines [15,168]. However, NF-κB-mediated regulation of autophagy is yet to be elucidated in kidney disease.

## 5. Cell Fate in Renal Injury and Regulation by Autophagy

### 5.1. Cell Fate in Renal Injury

Loss of tubular epithelial cells is a key feature of renal injury, and several modes of tubular cell death, including apoptosis and regulated necrosis/necroptosis, have been described; however, the underlying molecular pathways and their relationships remain to be determined.

#### 5.1.1. Apoptosis and Necrosis

Apoptosis, which was originally described in renal IRI in 1992 [203], has been subsequently demonstrated in IRI in many studies, including a study of human AKI [204]. Multiple recent reports have provided evidence that modes of cell death other than apoptosis may play a crucial role in renal injury. Conditional Bax- and Bak-KO mice have been shown to afford partial protection against IRI [205], suggesting multiple modes of cell death. Our understanding of the various modes of cell death in renal injury is likely limited by methodology. For example, DNA fragmentation determined by the TUNEL assay has been used as a hallmark of apoptosis in in vivo renal injury studies in animal models; however, the TUNEL assay cannot be considered as conclusive for apoptosis [206] because DNA fragmentation also occurs in necroptosis [207]. Also, the pan-caspase inhibitor zVAD does not provide protection against tubular damage or loss of renal function in renal IRI [208]. Nevertheless, in these studies, such results may be confounded due to apoptosis-independent effects of caspase inhibitors, including impaired autophagic flux and necrotic cell death [147,209].

Necrosis, which until recently was thought to occur in an accidental manner, is now recognized as a highly regulated pro-inflammatory process known as regulated necrosis, wherein the plasma membrane integrity is lost, and damage-associated molecular pattern proteins (DAMPs) are released [210,211]. Multiple forms of regulated necrotic pathways, including necroptosis, mitochondrial permeability transition pore (MPTP)-mediated necrosis, ferroptosis, parthanatos, and pyroptosis, have now been identified [211,212].

#### 5.1.2. Necroptosis

Necroptosis is triggered by diverse stimuli, including the well-studied death receptors of the tumor necrosis factor-alpha (TNF-α) superfamily [211,212]. Signals transduced upon activation of death receptors (i.e., TNFR1) by binding with their respective ligands (i.e., TNF- α) results in the formation of a downstream cytosolic complex of the necrosome-containing the receptor-interacting protein kinase (RIPK) 3- RIPK1 complex and mixed lineage kinase domain-like protein (MLKL) [213,214,215,216]. Phosphorylation of MLKL by RIPK3 promotes a conformational change and oligomerization of MLKL, which facilitates plasma membrane lysis to induce necroptosis (Figure 3A) [211,212,217,218]. The term ‘necroptosis” was designed for regulated necrotic cell death, wherein necrostin-1, a highly specific RIPK1 inhibitor, inhibits death receptor-induced necrotic cell death in the presence of caspase inhibitors [219]. However, caspase-independent cell death requiring the Fas-associated death domain (FADD) and RIPK1 was described before the phenomenon of necroptosis had been recognized [220]. 

Both pharmacological and genetic approaches have revealed cell death by necroptosis and its role in models of AKI and CKD. Necrostatin-1, a highly specific RIPK 1 inhibitor, has been shown to provide protection against tubular damage and loss of renal function in AKI caused by IR [208,221]**,** cisplatin [222,223], or radiocontrast agents [224]. Severe combined immunodeficiency (SCID)-beige mice subjected to IRI were also protected by nerostatin-1, indicating that this protection by nerostatin-1 is independent of T and natural killer cells [205]. Different forms of cell death, including apoptosis and regulated necrosis, are known to contribute to renal failure during IRI [34,204]. The inability of the pan-caspase inhibitor zVAD to protect against IR-induced tubular damage and renal function [208,224] suggests that apoptosis may not play a major role in IR-induced AKI. Indeed, necrostatin-1 was found to be nephroprotective in a mouse model of renal IR [208] and a rat model of CKD induced by a subtotal nephrectomy [225]. Moreover, *RIPK3*-KO mice were protected from IR-induced tubular damage and renal dysfunction [226], and both *RIPK3*-KO and *MLKL*-KO mice exhibited protection against cisplatin-induced necroptosis [223]. Although cisplatin is known to induce apoptosis, necrosis, and autophagy, the pan-caspase inhibitor zVAD did not protect against cisplatin-induced nephrotoxicity [147,223]. However, one study has reported that zVAD enhanced the protective effects of necrostatin-1 in cisplatin-induced AKI [227].

#### 5.1.3. Ferroptosis

Ferroptosis is another form of regulated necrosis that is characterized by iron-dependent reactive oxygen species (ROS) elevation that disrupts cellular redox homeostasis and augments lipid peroxidation, including peroxidation of polyunsaturated fatty acids [228,229]. It has been demonstrated that a lack of glutathione peroxidase 4 activity, which protects cells from lipid oxidation, leads to renal failure [230]. In addition, inhibition of ferroptosis by treatment with ferrostatin (a.k.a.16-86) was shown to protect mice from loss of renal function and structural tubular damage following IRI [231]. The ferrostatin effect was highly significant and exceeded the effects of other inhibitors of regulated necrosis previously found to confer protection against renal IRI [228]. A recent study also has revealed a synergistic effect of ferroptosis and necroptosis on tubular damage in IR injury and that both of these pathways interact to contribute to cell death [232]. 

#### 5.1.4. Mitochondrial Permeability Transition (MPT)-Regulated Necrosis (MPT-RN) 

MPT-RN is another form of regulated necrosis triggered by a sudden opening of the permeability transition pore complex in response to elevated cytosolic Ca^2+^ or ROS levels. The precise composition of the MPT pore is not known, but cyclophilin D has been shown to play a critical role in the opening of the pore. Mice lacking cyclophilin D are protected from both IRI [225,233] and cisplatin nephrotoxicity [226]. Related to MPT-RN, pathanatos is another cell death pathway of regulated necrosis that is dependent on the nuclear enzyme poly-(ADP-ribose) polymerase-1 (PARP-1). Hyperactivation of PARP-1 results in NAD depletion and disrupts the glycolytic pathway via hexokinase inhibition. Pharmacological inhibition or genetic deletion of PARP-1 is renoprotective under IR [233,234,235], cisplatin nephrotoxicity [236], diabetes [237], and ureteral obstruction [238] conditions.

### 5.2. Molecular Interaction of Autophagy with Apoptosis and Regulated Necrosis

Increased autophagy has been demonstrated to provide protection against renal tubular cell death in kidney injury [239] and podocyte cell death in lupus nephritis [240] and diabetic nephropathy [241]. At present, the mechanisms underlying the role of autophagy in apoptosis and regulated necrosis of renal tubular epithelial cell injury are not completely understood. Autophagy may provide a cytoprotective function without affecting a specific cell death pathway directly. It could be that AKI-associated metabolic stresses deprive cells of vital bioenergetic molecules and nutrients, which autophagy replenishes by degrading damaged macromolecules and organelles, thus generating the necessary amino acids and bioenergetic molecules needed to rescue otherwise dying cells. Conversely, it could be that the autophagy pathway may also participate and interact with the molecular machinery of a particular cell death pathway and influence the overall cell-fate. The cross-talk between autophagy and other modes of cell death is complex but may be critical in determining the overall cell fate as described below.

#### 5.2.1. Autophagy and Apoptosis

Autophagy can regulate apoptosis following molecular interactions between the key proteins of these pathways, including members of the Bcl-2 family, autophagy proteins, and caspases [27]. Beclin-1, through its Bcl-2 homology domain, binds to Bcl-2 and inhibits autophagy under normal or nutrient-enriched conditions [157]. The Bcl-2 homology domain, containing Puma, Noxa, Nix, Bid, and Bnip3 proteins, disrupts the association between the Bcl-2 proteins (Bcl-2, BclxL, and Mcl-1) and beclin-1, resulting in autophagy-promoting free beclin-1 [157]. Phosphorylation of Bcl-2 by JNK1 dissociates the beclin-1-Bcl2 complex under starvation conditions. and free beclin-1 then promotes autophagy [158]. Prolonged nutrient deprivation or starvation leads to increased Bcl-2 phosphorylation, which promotes the dissociation of Bax/Bak from Bcl-2 to facilitate cytochrome c release and subsequent apoptosis [242]. 

Molecular interaction studies of autophagy and apoptosis have shown that some key autophagy proteins, including beclin-1 [243,244], VPS34 [244], Atg3 [245], Atg4 [246], Atg5 [247], Atg16L [248], and AMBRA1 [249], are targeted by caspase-mediated cleavage, leading to suppression of autophagy. Also, after cleavage, the beclin-1 and Atg4 fragments translocate to the mitochondria, facilitating cytochrome c release and subsequent apoptosis [244,246]. In an AKI model of cisplatin nephrotoxicity, beclin-1, Atg5, and Atg12 were cleaved and degraded during injury, and the pan-caspase inhibitor zVAD-fmk prevented cleavage of autophagy proteins [147]. Moreover, the degradation of autophagy proteins has been attributed to reduced autophagy during cisplatin injury. Thus, selective degradation of damaged mitochondria by mitophagy can reduce apoptosis [250]. 

#### 5.2.2. Autophagy and Necroptosis

Necroptosis can be suppressed or promoted by autophagy, depending on the context and cell types. However, the molecular relationship and the underlying mechanisms between autophagy and necroptosis are not well understood. Autophagy has been shown to inhibit TNFα- and zVAD-induced necroptosis in mouse L929 fibrosarcoma cells [251,252]. Similarly, in tumor cells, autophagy has been demonstrated to provide protective function against necrotic cell death, and suppression of both apoptosis and autophagy promoted necrotic cell death [253]. In addition, KU55933, a small-molecule inhibitor of autophagy, induced necroptosis and apoptosis in amino acid-starved MCF7 human breast carcinoma cells, demonstrating that autophagy protects starved cells against both apoptosis and necroptosis [254]. In renal carcinoma cells, autophagy induced by mTOR inhibition (by CCI-779) or accumulation of autophagosomes (by chloroquine) resulted in RIPK3-dependent necroptosis [255]. Moreover, TNF-α and z-VAD-fmk treatments induced necroptosis and impaired autophagic flux in cardiomyocyte H9c2 cells, while increasing RIPK1-p62 binding and reducing p62-LC3 binding; meanwhile, rapamycin treatment partly restored autophagic flux and suppressed TNF/zVAD-induced necroptosis [256]. Furthermore, cellular FLICE (FADD-like IL-1β converting enzyme) inhibitory protein (cFLIP) regulates necroptosis by forming a so-called ‘ripoptosome’, a death receptor-independent cell death complex containing RIPK1, FADD, and caspase-8 (Figure 3A). While cFLIP promotes riptosome assembly and subsequent necroptosis by inhibiting caspase-8, cFLIPL prevents its assembly by activation of caspase 8, which then cleaves RIPK1 in the complex [257]. cFLIPL can suppress autophagy by preventing Atg3 from converting LC3 to LC3-II [258,259] (Figure 3B). Upon adhesion, isolated blood eosinophils are lysed by necroptosis, and induction of autophagy by rapamycin counter regulates the eosinophil cytolysis [260], suggesting that activation of autophagy prevents adhesion-induced eosinophil cytolysis. Autophagy also provided protection against necroptosis in DU145, a human prostate cancer cells [261], and in malignant pleural mesothelioma cells [261]. In A549 human lung cancer cells, necroptosis induced by the naturally occurring naphthoquinone shikonin was significantly enhanced by inhibition of autophagy [262]. Additionally, in a model of status-epilepticus-induced hippocampal damage, curcumin-induced autophagy reduced neuronal necroptosis [263].

The molecular relationship between autophagy and regulated necrosis has not been determined in renal injury. In non-renal cells, necroptosis has also been shown to activate autophagy, as depicted in Figure 3A,B. For example, in endothelial cells, inhibition of autophagy rescued palmitic acid-induced necroptosis of endothelial cells [264], and autophagy-mediated necroptosis was required to overcome glucocorticoid resistance in childhood acute lymphoblastic leukemia cells [265]. Moreover, hepatic I/R increased RIPK1/RIPK3 necrosome formation, as well as activation of autophagy and mitophagy. Autophagy suppression by necrostatin-1 mediated inhibition of necroptosis, suggesting that necroptosis may induce autophagy during hepatic IR [266]. In rhabdomyosarcoma cells, autophagy stimulated by the indole bipyrrole compound GX15-070 promoted the assembly of the necrosome complex that elicited necroptosis [267]. Also, in mouse prostate epithelial cells, the autophagy machinery was demonstrated to provide a scaffold for efficient necrosome activation by phosphorylation of MLKL to facilitate cells to undergo necroptosis [268]. In this process, autophagy-mediated activation of necrosomes was caused by p62/SQSTM1 recruitment of RIPK1. 

Finally, genetic and pharmacological inhibition of BMI1, a proto-oncogene, induced autophagy-mediated necroptosis in ovarian cancer cells [269].

#### 5.2.3. Autophagy and Ferroptosis

In fibroblasts and cancer cells, activation of the autophagy promotes ferroptosis by the degradation of ferritin [26]. Erastin-induced ferroptosis has been shown to be suppressed by *Atg5* or *Atg7* deficient mouse embryonic fibroblasts (MEFs), as well as in human pancreatic cancer and human fibrosarcoma cell lines with knockdown of Atg5 or Atg7 protein levels with shRNA [26]. Furthermore, nuclear receptor coactivator 4 (NCOA4), a cargo receptor for autophagic turnover of ferritin [270,271], also has been shown to be involved in erastin-induced ferroptosis [26].

#### 5.2.4. Autophagy and PARP-Mediated Necrosis 

As described above, PARP expression and activity are increased in AKI [272]. Elevated PARP activation results in rapid depletion of intracellular ATP due to NAD^+^ consumption and subsequent necrotic cell death [273]. Therefore, activation of AMPK due to ATP depletion may promote the induction of autophagy. In support of this possibility, it is known that inhibition of PARP-1 decreases AMPK levels and autophagy [274,275]. In addition, autophagy induction plays a protective role in PARP-mediated necrotic cell death [276,277]. In CNE-2 human nasopharyngeal carcinoma cells, PARP-1 promotes autophagy via the AMPK/mTOR pathway in response to ionizing radiation, and inhibition of autophagy contributes to the radiation sensitization of CNE-2 cells [278].

## 6. Concluding Remarks

Studies in conditional proximal tubule-specific autophagy-KO mice (tissue-specific Atg5- or Atg7-KO mice) have consistently demonstrated worsened outcomes in response to renal injury, pointing towards a pro-survival role of autophagy. To determine a more direct effect of autophagy in renal injury, further studies are required using specific genetic procedures that stimulate autophagy in renal tubules. This can be accomplished either by generating a proximal tubule-specific transgenic mouse that overexpresses autophagy in tubules or by using specific inducers of autophagy that function without their side effects. Although some studies have examined the role of autophagy in progressive renal disease, additional studies with different experimental models of renal fibrosis with both genetic and pharmacological approaches are required to determine the definitive role of autophagy in renal interstitial fibrosis. Commonly used mTORC1 inhibitors are not clinically relevant inducers of autophagy and have adverse effects because mTORC1 is a multifunctional serine-threonine kinase that is vital for maintaining tubular homeostasis. Recently documented mTORC1-independent alternative pathways may contribute to the autophagy activation, and investigation of these pathways in renal injury may provide useful information on stimulating autophagy independent of mTORC1. Therefore, there is an urgent need to develop more effective and specific pharmacological inducers of autophagy that will provide beneficial effects in renal injury. A good example of such specific inducers of autophagy is beclin-1-derived TAT-beclin-1 autophagy-inducing peptides, which can be tested as an enhancer of autophagy during renal injury. Recent studies have begun to unfold the role and regulation of autophagy in renal injury; however, there is a paucity of information on elucidating the specific defects in autophagy activation and completion of the process of autophagy flux during renal injury. A better understanding of the mechanisms of autophagy dysregulation in renal injury will help to discover ways to overcome specific defects. To determine the specific role of autophagy in renal interstitial fibrosis, it is critical to examine the role of autophagy in matrix deposition by stimulating as well as by deleting autophagy not only in tubular cells but also in specific matrix-forming cells, including myofibroblasts and their precursor cells. At present, it is not known how autophagy influences different modes of cell death in renal injury and what fundamental molecular interactions occur in a dynamic balance to determine the overall cell fate. If autophagy induction plays a pro-survival role, future studies are needed to examine how autophagy promotes the survival of tubular cells undergoing apoptosis and regulated necrosis during renal injury.

## Figures and Tables

**Figure 1 biomolecules-10-00100-f001:**
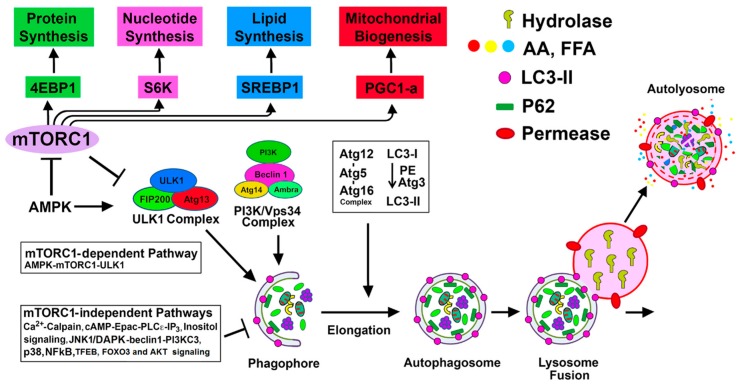
mTORC1-dependent and mTORC1-independent regulation of autophagy activation: Suppression of mTORC1 and induction of adenosine monophosphate activated protein kinase (AMPK) promote Unc-51 like autophagy activating kinase (ULK1) complex [ULK1, Atg13, focal adhesion kinase family interacting protein of 200 kD (FIP200), and Atg101] activation at the pre-autophagosomal assembly site (the certain domain of the ER) and initiates the autophagy process. Active ULK1 complex regulates the activity of the class III phosphatidylinositol (PtdIns) 3-kinase complex (including Beclin-1, Atg14(L)/barkor, Vps15, Vps34, and Ambra1) that generates phosphatidylinositol 3-phosphate (PI3P) rich domain. PI3P and PI3P-binding proteins (double FYVE domain-containing protein 1 (DFCP-1) and WIPI proteins) participate in the nucleation and associated membrane dynamics of the phagophore structure, the site of nucleation. Atg9 positive vesicles, that traffic from Golgi and endosomes and regulated by ULK1 complex, contribute to the formation of omegosomes and phagophores. Expansion and maturation of autophagosome from the phagophore structure require two ubiquitin-like conjugation systems that produce Atg12-Atg5-Atg16 oligomeric complex (Atg16 L1 complex) and lipidation of microtubule-associated protein 1A/1B-light chain 3 (LC3) (mammalian homolog of yeast Atg8) with phosphatidyl-ethanolamine (PE). WIPI upon binding to PI3P recruits Atg16 L1 complex to PI3P initiation sites. Atg12-Atg5 of the Atg L1 complex is then involved as an E3-like enzyme for the formation of the lipidated form of LC3. Once the autophagosome is produced, it then fuses with the lysosome to form autolysosome, and subsequently, the lysosomal hydrolases degrade the sequestered cargo. mTORC1 positively regulates protein synthesis, nucleotide synthesis, lipid synthesis, and mitochondrial biogenesis, as shown in the figure. mTORC1-independent alternative pathways negatively regulate autophagy.

**Figure 3 biomolecules-10-00100-f003:**
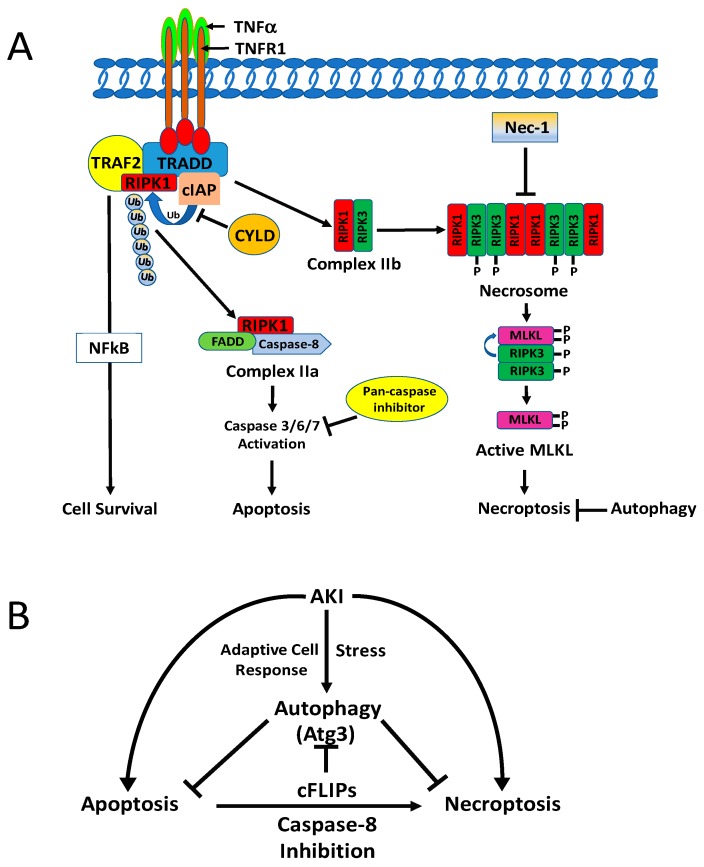
(**A**) Binding of tumor necrosis factor α (TNFα) with TNF receptor activates the receptor, which recruits TNFR1-associated death domain protein (TRADD). TRADD can recruit several binding partners, including shown in Table 2. Receptor-interacting serine/threonine-protein kinase 1 (RIPK1) and inhibitor of apoptosis proteins (cIAP) are involved in different downstream signaling pathways, including NF-κB activation, apoptosis, and necroptosis. TNFR-associated factor 2 (TRAF 2) binds to its cytoplasmic death domain. TRADD has several protein-binding partners and participates in different signaling pathways, including NF-κB, apoptosis, necrosis, and mitogen-activated protein (MAP) kinase activation. The N-terminal domain of TRADD interacts with the C-terminal domain of TRAF2 and recruits TRAF2 to TNFR1 for activation of the NF-κB pathway. E3 ubiquitin ligases cIAP1/cIAP2 polyubiquitinate (Ub–Ub) RIP1 that promotes the recruitment of inhibitor of nuclear factor kappa-B kinase (IKK) complex and TAK1, for NF-κB activation. In the absence of cIAP1 or deubiquitination of RIPK1 by cylindromatosis (CYLD), a K63-specific deubiquitinating enzyme (DUB) mediates the deubiquitination of RIP1 to facilitate the formation of complex II (IIa and Iib), which is dissociated from the receptor when TNFR is internalized. cFLIP, RIP1, FADD, and caspase-8 form cytosolic complex IIa to activate the caspase cascade and induce apoptosis. Active caspase 8 also cleaves and inactivates RIPK1 and RIPK2. When a caspase-8 activity is compromised by cFLIPs, vFLIP, or zVAD fmk, RIPK1 interacts with RIPK3 in a “necrosome” complex. RIP3 upon activation phosphorylates MLKL, and this promotes oligomerization of MLKL, resulting in its insertion in the plasma membrane to execute necroptosis. (**B**) Molecular interactions between necroptosis and autophagy show the involvement of cellular FLICE (FADD-like IL-1β converting enzyme) inhibitory protein (cFLIP), caspase-8, and Atg3. cFLIPs promotes necrosome assembly and subsequent necroptosis by inhibiting caspase-8; cFLIPL prevents its assembly by activation of caspase 8 that cleave RIP1 in the complex. cFLIPL can suppress autophagy by preventing Atg3 from converting LC3 to LC3-II.

**Table 1 biomolecules-10-00100-t001:** Autophagy in renal interstitial fibrosis.

Kidney Disease	Agent/Drug	Effect on Autophagy	Effect on Fibrosis	Reference
Autophagy suppresses fibrosis				
UUO model	3-MA	↓ autophagy	↑ in interstitial fibrosis and tubular apoptosis	[42,43,44]
	LC3 KO and beclin-1 ±	↓ autophagy	↑ deposition of collagen and TGF-β1	[46]
	Conditional deletion of ATG7 in distal tubule	↓ autophagy in the distal tubules	↑ in tubulointerstitial fibrosis via the TGF-β/Smad4 and NLRP3 signaling	[47]
	Conditional deletion of ATG5	↓ autophagy	↑ renal interstitial fibrosis and cell cycle arrest at G2/M	[48]
	Proximal tubule specific deletion of ATG5	↓ autophagy	↑ renal fibrosis due to leukocyte infiltration and expression of pro-inflammatory cytokines	[49]
	Valproic acid (histone deacetylase inhibitor)	↑ autophagy	↓ in renal fibrosis	[51]
	Rubicon	↓ autophagy	↑ in renal fibrosis	[52]
STZ- DN	miR -22 upregulation	↓ autophagy	↑ in renal fibrosis with increased expression of col-IV and α-SMA	[53]
	Triptolide	↑ autophagy via miR-141-3p/PTEN/Akt/mTOR pathway	↓ in renal fibrosis	[54]
HFD with CKD	Elafibranor (dual PPAR*α*/*δ* agonist)	↑ autophagy mediated by SIRT1	↓ in renal fibrosis	[56]
5/6-Nephrectomy	Knockdown of periostin gene (osteoblast specific factor-2)	↑ autophagy and upregulates periostin gene (pro-fibrotic and pro-inflammatory factor)	↓ in renal inflammation and fibrosis	[58]
	Proximal tubule specific deletion of ATG7	↑ autophagy	↓ in renal fibrosis with pro-fibrotic FGF2	[60]
Autophagy promotes fibrosis				
Ang II- induced CKD	1,25-dihydroxyvitamin D3	↓ autophagy with improved mitochondrial dysfunction	↓ in renal fibrosis	[55]
Stage 3 and stage 4 CKD	Rhubarb (Rhein- bioactive component)	↓ autophagy	↓ in renal fibrosis	[61]

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
