# Peer review of "Autophagy Function and Regulation in Kidney Disease"

_biomolecules, 2020, doi:10.3390/biom10010100_

Round 1

Reviewer 1 Report

The review is very interesting since autophagy is implicated, mainly as a protective mechanism, in many, if not all, kidney diseases. The authors evaluated the literature extensively and they provide data about the molecular mechanisms that govern autophagy, its role in various models of kidney disease, the pathways that induce autophagy, which are possible therapeutic targets, and finally the association of autophagy with various types of cell death. The manuscript is well-written, and the figures well-designed and informative.

I have the following comments.

Another, not mentioned in the review mechanism that induces autophagy and protects primary human renal tubular epithelial cells (RPTECs) subjected to hypoxia against apoptosis is the activation of general nonderepressible-2 (GCN2) kinase. Amino acid deprivation activates GCN2 kinase which increases autophagy, possibly through a p53-mediated overexpression of BNIP3L. In parallel GCN2 kinase activation increases cellular ATP content and inhibits apoptosis assessed by cleaved caspase-3. P21 is upregulated, whereas Bax is downregulated (Eleftheriadis et al., Int Urol Nephrol 2017; 49:1279). Such a mechanism may be implicated in the protective effect of relatively low-protein diets on the progression of chronic kidney disease. Interestingly, and contrary to mTORC1, which is inhibited by deprivation of certain amino acids, and more precisely of leucine, isoleucine, valine and possibly arginine, GCN2 kinase is activated by deprivation of any amino acid (Gallinetti et al., Biochem J 2013; 449:1). Other studies, has also confirmed the role of GCN2 kinase activation in inducing autophagy (B’Chir et al., Nucleic Acids Res 2013; 41:7683 / Fougeray et al., J Immunol 2012; 189:2954). Notably, from all amino acids, tryptophan depletion may take place easily through a 2-day low protein diet (Laeger et al., J Clin Invest 2014; 124:3913, Ellenbogen et al., Neuropsychopharmacology 1996; 15:465) or due to its degradation by indoleamine 2,3-dioxygenase-1 (IDO-1) under inflammatory conditions.  Increasing kidneyIDO1 activity or treating mice with a GCN2 agonist induces autophagy and protects mice from nephritic kidney damage Chaudhary et al., J Immunol. 2015;194:5713). It should be noted that I-R injury consists of two subsequent, but distinct components, the ischemia, and the reperfusion. When these two components are studied separately, some impressive results are obtained regarding autophagy, apoptosis, and types of programmed cell necrosis. In experiments with isolated mouse renal tubules, I-R induces ferroptosis but not necroptosis. However, the same group of investigators found that in vivo necroptosis prevails in the pathogenesis of I-R-induced AKI. Hence, it is likely that ferroptosis of RPTECs may be the initial insult, and the subsequent release of danger-associated molecular patterns and recruitment of immune cells trigger a second wave of cell death through necroptosis (Linkermann et al, Kidney Int. 2012; 81: 751 & Proc. Natl. Acad. Sci. USA 2014; 111: 16836). Thus, interfering with the initial insult, which is ferroptosis, may be extremely useful from a clinical point of view. Also, in cultured of primary human RPTECs, conditions that simulate ischemia induce apoptosis and autophagy, whereas, during the subsequent reoxygenation, autophagy, apoptosis, and ferroptosis take place. Necroptosis is not detected either under anoxia or reoxygenation (Eleftheriadis et al., Biology 2018; 7, 48; DOI:10.3390/biology7040048).

Author Response

We thank the reviewer for critical evaluation of the review.

As suggested we have added the information on GCN2 -mediated autophagy induction inkidney diesease in section 4.2.9.

Reviewer 2 Report

In this focused review, the authors' highlight the current knowledge base on the role of autophagy in the context of kidney diseases. Although the focus is very specific, the material is presented in logical and readable fashion, and the tables are adequately used to summarize the available literature.

Author Response

We thank the reviewer for critical evaluation of the review

Round 2

Reviewer 1 Report

The authors addressed the raised issues adequately.